# Metabolic and Productive Response and Grazing Behavior of Lactating Dairy Cows Supplemented with High Moisture Maize or Cracked Wheat Grazing at Two Herbage Allowances in Spring

**DOI:** 10.3390/ani11040919

**Published:** 2021-03-24

**Authors:** Verónica M. Merino, Lorena Leichtle, Oscar A. Balocchi, Francisco Lanuza, Julián Parga, Rémy Delagarde, Miguel Ruiz-Albarrán, M. Jordana Rivero, Rubén G. Pulido

**Affiliations:** 1Departamento de Producción Animal, Facultad de Agronomía, Universidad de Concepción, P.O. Box 160-C, Concepción 4030000, Chile; veronicamerino@udec.cl; 2Escuela de Graduados, Facultad de Ciencias Agrarias, Universidad Austral de Chile, P.O. Box 567, Valdivia 5090000, Chile; loreleichtle@hotmail.com; 3Facultad de Ciencias Agrarias, Instituto de Producción Animal, Universidad Austral de Chile, P.O. Box 567, Valdivia 5090000, Chile; obalocch@uach.cl; 4Private Consultant, Osorno 5290000, Chile; franciscolanuza@gmail.com (F.L.); jmpargam@gmail.com (J.P.); 5PEGASE, INRAE, Institut Agro, 16 Le Clos, F-35590 Saint-Gilles, France; remy.delagarde@inrae.fr; 6Faculty of Veterinary Medicine and Animal Science, Autonomous University of Tamaulipas, Victoria 87049, Mexico; miguel.ruiz@docentes.uat.edu.mx; 7Departamento de Ciencias Agropecuarias y Acuícolas, Facultad de Recursos Naturales, Universidad Católica de Temuco, Temuco 4780000, Chile; jordana.rivero-viera@rothamsted.ac.uk; 8Rothamsted Research, North Wyke, Okehampton, Devon EX20 2SB, UK; 9Instituto Ciencia Animal, Facultad de Ciencias Veterinarias, Universidad Austral de Chile, P.O. Box 567, Valdivia 5090000, Chile

**Keywords:** energy supplementation, grazing management, milk production, metabolic response, ruminal function

## Abstract

**Simple Summary:**

Energy supplements such as high moisture maize or cracked wheat increase total dry matter intake (DMI) and dairy cow performance compared to pasture-only diets. However, the effectiveness of such a feeding strategy depends upon the level of herbage allowance (HA). In this study, increasing HA from 20 to 30 kg DM/cow had no effect on milk production but increased the concentration of urea in milk and plasma regardless of the type of energy supplement offered to grazing dairy cows. These results demonstrate that in high-quality pasture, low HA is appropriate to improve milk production performance per cow and per hectare.

**Abstract:**

The aim was to determine the effect of the herbage allowance (HA) and supplement type (ST) on dry matter intake (DMI), milk production and composition, grazing behavior, rumen function, and blood metabolites of grazing dairy cows in the spring season. Experiment I: 64 Holstein Friesian dairy cows were distributed in a factorial design that tested two levels of daily HA (20 and 30 kg of dry matter (DM) per cow) and two ST (high moisture maize (HMM) and cracked wheat (CW)) distributed in two daily rations (3.5 kg DM/cow/day). Experiment II: four mid-lactation rumen cannulated cows, supplemented with either HMM or CW and managed with the two HAs, were distributed in a Latin square design of 4 × 4, for four 14-d periods to assess ruminal fermentation parameters. HA had no effect on milk production (averaging 23.6 kg/day) or milk fat and protein production (823 g/day and 800 g/day, respectively). Cows supplemented with CW had greater protein concentration (+1.2 g/kg). Herbage DMI averaged 14.17 kg DM/cow.day and total DMI averaged 17.67 kg DM/cow.day and did not differ between treatments. Grazing behavior activities (grazing, rumination, and idling times) and body condition score (BCS) were not affected by HA or ST. Milk and plasma urea concentration increased under the high HA (+0.68 mmol/L and +0.90 mmol/L, respectively). Cows supplemented with HMM had lower milk and plasma urea concentrations (0.72 mmol/L and 0.76 mmol/L less, respectively) and tended (*p* = 0.054) to have higher plasma β-hydroxybutyrate. Ruminal parameters did not differ between treatments.

## 1. Introduction

In temperate regions, such as New Zealand, Australia, Ireland, western France, and southern Chile, grazed pastures are the main and cheapest source of nutrient for dairy cows [1], where a higher proportion of grazed pasture in the annual diet improves the economic efficiency achieved on dairy farms [2]. Pasture-based dairy systems require the use of grazing managements that ensure an adequate provision of high-quality herbage plus a strategic supplementation with low-cost feeds to support the increased milk production and milk solids per hectare [3]. The intensification of these systems through the use of supplements to increase DMI and milk yield per cow can reduce the profitability of grazing dairy systems if the cost of the marginal milk produced is greater than the additional milk price received [4]. In order to ensure the long-term profitability of pasture-based production systems, farmers and scientists are interested in finding strategies that allow balancing a high DMI per cow and high efficiency of pasture utilization [5].

Animal performance in grazing systems is frequently limited by the inability to consume enough herbage dry matter (DM) and energy to meet the nutrient requirements for higher milk production [6] and use N efficiently [7]. Therefore, farmers have been increasingly using feeding systems that combine pasture and feed supplements such as concentrates, grain, and conserved forage [8]. Moreover, the large seasonal variations in herbage production and quality, which are related to the growth stage, plant nutrition, frequency, and intensity of pasture use [9], generate additional challenges to farmers especially in early-spring when the pasture is generally in a vegetative state with high crude protein (CP) concentration [10] and cows are in the transition period usually with a negative energy balance [11].

The main objective of supplementation of grazing dairy cows is to increase total DMI and energy intake relative to that achieved with pasture-only diets [1]. A recent review has stated that the main opportunities for increasing milk production per cow and per ha are as follows: (a) careful pasture management to ensure adequate provision of high-quality herbage and (b) strategic supplementation with low-cost, high-energy feeds [12]. The energy supplementation strategies include different combinations of level, type, and processing of supplemental feeds [13]. Nevertheless, the type and level of supplementation used is strongly dependent on the relationship between milk and supplement prices [4].

There are several sources of energy supplementation, with maize and wheat as the main energy source in diets of high-producing dairy cows because they are cost-effective sources of digestible energy [14] to increase milk production [6] and milk protein percentage [15]. Either of those cereal grains offer a very good source of energy for dairy cows that graze pastures, with metabolizable energy (ME) values ranging from 12.3 to 14.9 and from 10.5 to 14.1 MJ ME/kg DM for maize and wheat grain, respectively [16].

Productive responses of lactating dairy cows to wheat- and corn-based diets depend on the, the grain source, dietary inclusion level, the level of dietary intake, the physical processing of the grain [17], the stage of lactation [18], and the genetic potential of the dairy cow [19]. Cereal grain processing is intended to improve the ruminal and total availability of starch [20], offering the opportunity to respond to market forces and health recommendations as well as improving feed conversion efficiency [21]. Starch is the major nutrient providing energy from these cereal grains, with differences in their starch content (77% and 72% DM basis for wheat and maize, respectively) [22] and in their starch ruminal degradability (with 32%/h and 2%/h for wheat and maize starch, respectively) [23]. The slower starch degradation in maize [14] has been attributed to a higher proportion of peripheral and horny endosperm compared to wheat that increases the resistance to microbial activity [24]. Thus, improving starch utilization from maize processing methods may improve lactation performance and reduce feed costs, especially during periods of high grain prices [25].

High-moisture maize results in the break-down of the hydrophobic starch-protein matrix, and consequently increase rumen starch degradability [25,26], and therefore contributes to synchronize energy supply with the rumen-degradable protein available and to optimize N assimilation and rumen microbial protein synthesis [27]. However, its high ruminal starch digestion rate limits the amount that can be included in the diet in order to avoid the negative effects on DMI and milk yield caused by a decrease in ruminal pH with the consequent increased risk of metabolic disorders such as acute and subacute ruminal acidosis, laminitis and liver abscesses [28].

Herbage allowance, defined as the amount of herbage offered daily per cow (kg DM/cow/d), is widely recognized as the main grazing management factor to improve the efficiency of utilization of available forage [19,29] and milk output per hectare on pasture-based systems [30,31]. However, herbage DMI could be substituted by the supplement intake, affecting the milk response to these supplements and consequently the efficiency of the feeding strategy [32].

Several studies have focused on identifying the combined effects of HA and supplemental grain feeding on DMI and milk production per cow [29,31,33,34,35,36], and on the pasture production and composition in the long-term [37,38]. However, there is no consensus among these studies on the optimal level of HA to offer dairy cows daily [6,34], and the question that remains, when pasture constitutes a large proportion of the diet, what is the optimal energetic supplement type to be offered to lactating dairy cows? Understanding how the type of energy-source supplement and the level of herbage allowance influence the grazing behavior and DMI, and hence the metabolic and productive response, are key challenges for improving the profitability of graze-based dairy systems. The objective of this study was to compare animal and metabolic responses and grazing behavior of dairy cows supplemented with HMM or CW, grazing at two HAs during spring.

## 2. Materials and Methods

The study was carried out at the Vista Alegre Experimental Research Station of the Universidad Austral de Chile, Valdivia, Chile (39°47′46″ S and 73°14′13″ W), in spring (77-d period from October to December 2010) in addition to a 14-d pre-experimental period to adapt the animals to the experimental conditions. Daily temperature and rainfall were collected from a meteorological station (Geoscience Institute, University Austral of Chile, Valdivia, Chile). The climate of the zone has been classified as temperate with Mediterranean influences [39]. The soil is a moderately deep Andisol (Duric Hapludand, soil Serie Valdivia) according to CIREN [40] with free drainage.

### 2.1. Experiment I: Treatments and Experimental Design

Four treatments were compared in a 2 × 2 factorial design, with two HAs; high, 30 kg DM/cow.day, and low, 20 kg DM/cow/day, and two types of supplement; HMM or CW offered at a level of 3.5 kg DM/cow/day distributed in two equal rations during milking time (06:00 and 15:00). Sixty-four Holstein Friesian dairy cows, 32 in early lactation (days in milk (DIM): 59 ± 33 days; milk yield: 29 ± 5.7 kg/day; LW: 530 ± 63 kg; BCS: 2.7 ± 0.4 points and parity: 3.3 ± 1.5 calving) and 32 of late lactation (DIM: 206 ± 10 days; milk yield: 23 ± 4 kg/cow/day; LW: 544 ± 85 kg and BCS: 3.0 ± 0.4 points) were randomly assigned to one of four dietary treatments (eight cows of each lactation stage per treatment, i.e., 16 cows per treatment).

### 2.2. Grazing Management and Animal Feeding

Grazing took place on 27 ha of a 12-years-old pasture dominated by perennial ryegrass (*Lolium perenne* L.), that had been subjected to rotational management. Herds grazed at different paddocks classified as high (*n* = 13) or low (*n* = 9) HA from autumn 2008. Herds with the same HA grazed in the same paddock but separated by an electric fence according to the corresponding supplement type. All paddocks were grazed to a similar pre-grazing herbage mass during all the experimental period that ranged from 2200 to 2800 kg DM/ha. The grazing areas were calculated daily on the basis of pre-grazing herbage mass, estimated from 100 measurements made with a rising-plate meter (RPM, Ashgrove Plate Meter, Hamilton, New Zealand) and the corresponding level of HA assigned. The offered pasture area was subdivided into two strips each day. Measurements were made by walking the paddocks in a “W” pattern and this was repeated post-grazing, enabling grass disappearance of each individual herd to be calculated. All animals were given access to a new grazing strip after each milking.

The swards received a fertilization level of 52 kg P, 50 kg K, and 46 kg N in autumn, and 46 kg N in spring, based on a potential production level of 12 t DM/ha/y. The chemical composition of the herbage offered (>4 cm) was evaluated weekly during the experimental period (see Merino [37] for details).

The pastures grazed had a similar pre-grazing herbage mass (ground level) and compressed sward heights) averaging 2257 kg DM/ha and 9.3 cm, respectively [37]. The amount of pasture offered was provided by adjusting the offered area (89 vs. 134 m^2^/cow/day, for low and high HA treatments, respectively). The sward height post-grazing was 4.02 and 4.7 for low and high HAs, respectively, while the post-grazing herbage mass was 1203 and 1330 kg DM/ha for low and high HAs, respectively [37].

Mineral salts (ANASAL high production, ANASAC^®^: Ca 14.0%, P 10%, Mg 6%, Na 4%, S 0.2%, Zn 5000 mg/kg, Cu 1500 mg/kg, Co 20 mg/kg, and I 200 mg/kg) were offered with concentrate at rates of 0.25 kg/day per cow to prevent mineral deficiencies. Water was available freely in the pasture and in the holding area of the milking parlor.

### 2.3. Measurements and Samplings

Data of air temperature and rainfall during the 91-d period were collected from an automatic met station of the Institute of Geosciences of the Austral University of Chile (Isla Teja station) and compared to the historical precipitation and temperature record.

#### 2.3.1. Supplements Sampling

Once a week, five samples of HMM and CW were collected and dried for 48 h at 60 °C for chemical analysis. Supplements samples were ground through a 1-mm screen (Willey Mill, Arthur H Thomas Co., Philadelphia, PA, USA) and analyzed for DM and CP (according to the Association of Official Analytical Chemical [41]), and neutral detergent fiber (NDF, according to Van Soest [42]). Metabolizable energy was estimated by regression using a “D” value (digestible organic matter/DM × 100) and was determined in vitro [43] according to Goering and Van Soest [44].

#### 2.3.2. Behavioral Observations

Individual grazing behavior was recorded three times over 24-h periods (weeks four, seven, and nine). Observations were recorded by four trained observers for all animals every 10 min during daylight hours (06:00 to 19:59) and every 15 min at night (20:00 to 05:59). At each observation time, the activities of grazing, ruminating, and idling were recorded. For this purpose, cows were individually identified with a number painted in their flanks. Grazing time was evaluated as the amount of time (expressed in minutes) devoted to grazing, where cows were considered to be eating when bowing their heads down and consuming herbage [45].

The biting rate (bites/min) was assessed twice (weeks four and nine) in 15–17 cows each time on the same day as the behavior was recorded. The number of bites of each cow evaluated was recorded over two-minute periods with a hand-held counter to provide two measurements per day (around 30 min at the beginning of each grazing event, 06:30 and 15:30).

#### 2.3.3. Intake Calculations

Daily DMI of HMM and CW was determined from each cow by calculating the difference between the feed offered and refused. Voluntary herbage DMI was estimated for each cow in the experiment derived from the energy requirements of the cows and the ME content of the pasture and supplements as described by Baker [46] using the following equation:Herbage DMI (kg DM/day) = ((MME + MEL + MEiwc + MEG) − (supplement ME))/herbage ME(1)
where MME, MEL, MEiwc, and MEG are the requirements for maintenance, lactation, body weight change, and pregnancy, respectively. Supplement ME corresponds to the total ME consumed (cow/day) through the supplement, and herbage ME is ME per kg DM of herbage.

Total DMI was calculated by the daily herbage DMI plus the amount of concentrate intake. The herbage intake rate (g DM/min) was calculated as the ratio of herbage intake (kg DM/d) to grazing time (min/d).

#### 2.3.4. Milk Production, Milk Composition, Bodyweight, and Body Condition Score

Cows were milked at 06:00 and 15:00 and milk yield was registered at each milking time during the experiment. Representative subsamples were collected for two consecutive days on three occasions (weeks 3, 7, and 11 of the trial) at morning and afternoon milkings (these two subsamples were pooled within each sampling date) to determine milk fat, milk protein, and milk urea by infrared spectrophotometry (Milko-scan, system 4300 Foss Electric, Hilleroed, Denmark).

Cows were weighed once a week after morning milking using a Roman-type fixed scale in pens located adjacent to the milking parlor and the BCS was recorded by two experienced observers (their individual values were averaged) using the five-point scale [47].

#### 2.3.5. Blood Sampling Collection

Blood samples were obtained by venipuncture coccygeal with heparinized tubes from each cow after the afternoon milking on three occasions (weeks 4, 8, and 11 of the trial). Samples were centrifuged at 800× *g* for 10 min on the day of sampling, and the obtained plasma was aliquoted and frozen at −20 °C in 1.5 mL microtubes for later analysis. At the end of the trial, β-hydroxybutyrate (β-hydroxybutyrate Rambut, Randox^®^, Crumlin, UK), non-esterified fatty acids (NEFA, ACS-ACOD, Wako^®^, Neuss, Deutschland), albumin (BCG, Human®, Wiesbaden, Germany), and urea (GLDH, Human^®^) were determined in plasma in a Metrolab 2300 autoanalyzer (WienerLab^®^, Rosario, Argentina).

### 2.4. Experiment II: Treatments, Experimental Design, and Animals

Four ruminally cannulated multiparous Holstein–Friesian dairy cows were used to evaluate the treatment effects on ruminal fermentation variables. At the beginning of the experiment, the cows averaged 212 ± 72 DIM, 22.77 ± 4.5 kg of milk yield, 566 ± 59 kg of LW, and 3.0 ± 0.1 points of BCS. The cows were assigned to a 4 × 4 Latin square design with four 14-d periods. We used the same treatments described for Experiment I. The animals used the same pasture and grazed alongside the cows in the first experiment. Animal supplementation was the same as previously described.

### 2.5. Ruminal Measurements and Samplings

On day 14 of each experimental period, ruminal fluid samples were collected at 06:30, 11:00, 14:00, 16:30, 19:45, 22:30, and 00:45, to determine pH, NH_3_-N, and volatile fatty acid (VFA) concentrations. A representative ruminal fluid sample of the three compartments of the rumen (dorsal sac, ventral, and caudal) was taken; the content was squeezed and filtered to collect ruminal fluid. Immediately, the pH was measured in the sample obtained with a portable pH meter (HI 98127, Hanna Instruments Inc., Rhode Island, RI, USA). To determine total VFA concentration and the molar ratios of acetate, propionate, and butyrate, a volume of 20 mL was collected in tubes in duplicate, with 0.4 mL acidified with H_2_SO_4_ at 50% *v*/*v* frozen at −20 °C for subsequent analysis.

### 2.6. Laboratory Analysis

Ruminal samples were centrifuged at 3400× *g* for 5 min and the supernatant (10 mL) was transferred to 15 mL Falcon tubes which were centrifuged at 3400× *g* for 5 min. Then, 0.9 mL of the supernatant was transferred to a 1.5 mL microtube and 0.1 mL of formic acid was added, mixed, and centrifuged at 4200× *g* for 5 min [48]. The supernatant from each vial was transferred in duplicates to be subjected to gas-liquid chromatography (GLC) for analysis using GC (Shimadzu GC-2010) equipped with a flame ionization detector using a capillary column (BP21, 30 m × 0.32 mm, SGE) of acid-treated polyethylene nitroterephthalic. The carrier gas was helium with a column flow of 2.4 mL/min with a division ratio of the injector shown in 1:100. The temperature of the column oven program was 250 °C.

### 2.7. Statistical Analysis

In Experiment I, milk production (average value from week 3 to 11), milk fat and protein production, BCS (average value from week 3 to 11), and bodyweight (average value from week 3 to 11), were analyzed using analysis of co-variance by PROC Mixed SAS Institute [49] following the model below:Yijkl = µ + SLi + Pj + HAk + STl + HAk × STl + SLi × HAk + SLi × STl + b × Xijkl + eijkl
where Yijkl represents the analyzed variable, µ is the overall mean, SLi is the fixed effect of the lactation stage group (i = 1–2), Pj is the fixed effect of the parity (j = 1–2), HAk is the fixed effect of the herbage allowance (k = 1–2), STl is the fixed effect of the supplement type (l = 1–2), HAk × STl is the interaction between HA and ST, SLi × HAk is the interaction between lactation stage group and HA, SLi × STl is the interaction between lactation stage group and ST, b × Xijkl is linear effect of the pre-experimental centered covariate for each experimental variable (pre-experimental milk production was used as the covariate for milk production, milk fat production, and milk protein production), and eijkl is the residual error term. All other animal variables, for which covariates were not available (milk composition, blood metabolites, behaviors, and DM intakes), were analyzed using analysis of variance by PROC Mixed SAS [49] following the same model but without the effects of the covariates. The effect (*p*-values) of the lactation stage and the number of parity (primiparous or multiparous) are presented in the tables but the means are not shown (these factors were only included in the statistical model to account for the variation attributable to them).

In Experiment II, ruminal fermentation parameters were analyzed according to a 4 × 4 Latin square design using the GLM procedure of SAS [49] with the following model:Yijkl = µ + Pi + Cj +HAk + STl + HAk × STl + eijkl
where Yijkl, µ, Pi, Cj, HAk, STl, HAk × STl and eijkl represent the analyzed variable, the overall mean, the random effect of period (i = 1–4), the random effect of a cow (j = 1–4), the fixed effect of herbage allowance (k = 1–2), the fixed effect of supplement type (l = 1–2), the interaction between herbage allowance and supplement type, and the residual error term, respectively. For each cow and period, the variable value used for the analysis was calculated as an average of the seven values obtained from 00:45 to 20:30.

## 3. Results

### 3.1. Weather Conditions

Daily mean air temperature during the experimental period was 12.5 ± 2.1 °C (a minimum of 8.1 °C and a maximum of 18.7 °C). The total accumulated rainfall recorded during the study period was 410 mm.

### 3.2. Nutritive Value of Supplements

Table 1 shows the nutritional composition of the herbage (>4 cm) and supplements offered to grazing dairy cows during the experimental period. The chemical composition of the forage grazed was evaluated weekly during the experimental period, without differences between herbage allowances treatments, averaging 11.61 MJ/kg DM ME, 20.75% DM CP, and 39.7% DM NDF.

### 3.3. Grazing Behavior and Dry Matter Intake

Grazing time averaged 450 min/day and ruminating time averaged 487 min/day, with no difference between treatments (*p* > 0.05) (Table 2). The time spent in other activities, such as milking, drinking water, and idling was not different between treatments (*p* > 0.05) (Table 2).

Estimated herbage DMI averaged 14.17 kg DM/cow/day and total DMI averaged 17.67 kg DM/cow/day and did not differ between treatments (*p* > 0.05) (Table 2). Intake rate averaged 31.74 g DM/min and was not different among treatments (*p* > 0.05) (Table 2).

There was no significant effect of the HA nor ST on the bite rate (*p* > 0.05), however, there was a significant interaction between these main factors (*p* = 0.039). This interaction HA × ST resulted in no difference in bite rate under the low HA (averaging 63.2 and 62.1 ± 5.01 bite/min for CW and HMM, respectively) and a difference of 4.2 additional bite/min recorded on the cows supplemented with HMM compared with the cows supplemented with CW (65.2 vs. 61.0 bite/min) grazing at the high HA.

### 3.4. Animal Performance, Bodyweight, and Body Condition Score

Milk production averaged 23.5 kg/day and 4% FCM production averaged 21.8 kg/day and tended to increase with the low HA and supplementation of HMM (*p* = 0.088) (Table 3). Milk fat yield averaged 823 g/day and milk fat concentration was 35 g/kg and there were no differences between treatments (*p* > 0.05). Similarly, milk protein yield averaged 800 g/day and did not vary according to treatments (*p* > 0.05), but milk protein concentration was higher in cows supplemented with CW than in cows supplemented with HMM (*p* < 0.05). There was a tendency to have more protein concentration in the cows grazing a low HA and receiving CW (*p* = 0.0841, Table 3). Low HA increased the live weight of dairy cows (*p* < 0.05). BCS did not change between treatments (*p* > 0.05).

Low HA increased LW of dairy cows (*p* = 0.009). However, we found no significant differences in BCS among HA treatments (*p* > 0.05). Neither LW nor BCS were affected by the type of supplement, averaging 542 kg and 2.99 between LW and BCS, respectively. There was no significant interaction between HA and type of supplementation for LW or BCS.

### 3.5. Indicators of Energy and Protein Metabolism

Plasma concentration of β-hydroxybutyrate, NEFA, and albumin were not affected by HA nor supplement type (*p* > 0.05) (Table 3). Milk and plasma urea concentrations were greater (*p* < 0.05) in cows grazing a high HA, compared with a low HA. Cows supplemented with HMM had lower (*p* < 0.05) milk and plasma urea concentrations and tended (*p* = 0.054) to have higher plasma concentrations of β-hydroxybutyrate compared with CW.

### 3.6. Ruminal Fermentation

Herbage allowance and ST did not affect pH, NH_3_-N, or VFA ruminal concentrations (*p* > 0.05, Table 4). The pH presented a maximum value (6.4) at 06:30 and descended progressively to the lowest value around 22:00 (Figure 1a). The lowest pH (5.28) for each supplement was observed 8 h after supplementation (22:30, *p* < 0.05). Ruminal NH_3_-N concentrations remained stable during the morning but increased abruptly in the afternoon (*p* < 0.05, Figure 1b) reaching a maximum value at 19:45, and then descending to similar concentrations as in the morning (*p* > 0.05).

No difference was observed in the total VFA concentration (93.4 mmol/L) between treatments (*p* > 0.05, Figure 1c). Molar proportions of acetate, propionate, and butyrate averaged 62.1%, 21.3%, and 12.6%, respectively, and the acetate: propionate ratio averaged 3.03 and was not different between treatments (*p* > 0.05) (Table 4).

## 4. Discussion

Average daily temperatures during the 91-d period (pre-experimental and experimental time) were similar to previous years with an average of 13.1 °C (a minimum of 8.1 °C and a maximum of 18.7 °C). However, there was 23% less rainfall during spring 2010 than the historical average (40 years), especially in September [37].

### 4.1. Dry Matter Intake and Grazing Behavior

In strip and rotational grazing management, intake and grazing behavior are mainly affected by HA, described as the product between herbage mass and daily offered area [34,50]. Based on the predictive equations developed by Pérez-Prieto and Delagarde [30] through literature review and meta-analysis, the herbage DMI response to the increase in the level of HA (for the typical range of 20 to 40 kg DM/ha, measured above ground level) is 0.21 kg per kg of increase in HA. However, in the current study, the estimated herbage DMI and total DMI did not differ between HA treatments, which may suggest that the levels of HA tested may not have been sufficiently contrasting to impact on herbage DMI. However, the interpretation should be done with caution because when the herbage intake was estimated as the herbage mass pre-grazing using the rising plate meter technique, the individual herbage intake was increased by 2.9 kg/d in the low HA compared to the high daily HA treatment (*p* < 0.001) [37]. 

Cows offered the 20-kg HA had consistently lower offered area (45 m^2^/cow/day less) in each grazing event than those offered the 30-kg HA, which resulted in a greater grazing intensity as evidenced by the lower post-grazing herbage mass (127 kg DM/ha less) and post-grazing sward height (0.7 cm less) reported by Merino et al. [37], at similar pre-grazing herbage mass averaging 2257 kg DM/ha between HA treatments. The sward bulk density was much greater in the lower than in the upper strata [37,51]. However, this compensatory mechanism to avoid reductions in DMI and to sustain the levels of milk production [6] did not affect the grazing time, probably because their limit to compensate for a reduction in herbage intake is determined for the time required for other activities such as ruminating [52]. Significant differences in DMI between HA may be expected in unsupplemented dairy cows. When dairy cows are fed with concentrate, cows generally substitute some of the pasture that would have been eaten by the supplement [53]. The decrease in herbage DM intake when grazing dairy cows are fed supplements have been previously reported with corn silage [54,55] and with cereal-based concentrate [33] and it is expected to be higher for supplemented cows grazing at high HA [33,56].

The lack of effect of HA on herbage intake was in line with the absence of effect on all grazing behavior variables. The similar herbage intake rate and ruminating time observed between HAs in our study can be explained by the similar pre-grazing condition in terms of herbage mass, sward height, species density, and botanical composition reported by Merino et al. [37]. The weight of herbage consumed in each bite is constrained by the mass of plant material within the bite horizon particularly by the green leaf mass [57]. The sward condition changes rapidly as grazing progresses, decreasing the ratio of leaf to stem [3,58] and, therefore, the intake rate [59]. The dynamics of pasture depletion and changes of all morphological components of pasture (proportion of sheath, stem, and dead material) as the cows grazed progressively down through the pasture were evaluated by Merino et al. [37] without differences between HAs. Thus, the lack of extension of daily grazing time to compensate for low herbage availability under the low HA condition may be because the motivation to continue grazing down (below 5 cm of height) could be decreased due to the difficulty for grazing caused by a higher proportion of stems and dead material in the lower strata into the sward profile (averaging 19.9 and 13.2% between HAs, respectively) [37], limiting the grazing depth and grazing time [59,60].

Concentrate type had no effect on herbage DM intake nor on grazing behavior variables, probably because of the moderate and equal level of supplementation that cows received (3.5 kg DM/cow/day) [1] and because their similar energy content, which possibly resulted in a similar energy intake, being consistent with the reports of Pulido et al. [61] and Sayers et al. [62]. Greater differences may be found when cows are supplemented with a fiber-based concentrate than when cows are supplemented with a starch-based concentrate [63]. A further reason for the lack of effect of concentrate could be the low genetic merit of the cows used, as well as the fact that 50% of cows were in late lactation.

The increased bite rate observed immediately after milking on cows grazing the high HA and supplemented with HMM (compared with the cows grazing under the same HA but supplemented with CW), could be attributed to the differences in DM concentrations between supplements (692 vs. 840 g/kg DM for HMM and CW, respectively) [64], and by the slower ruminal degradability of HMM compared to CW, and lower energy supply after each milking. The lack of difference in bite rate between ST recorded in cows grazing at a low HA could be due to there being a greater grazing pressure that incentivizes herbage intake.

### 4.2. Milk Production and Composition, Bodyweight, and Body Condition Score

The effects of daily HA on dairy cow performance under spring grazing conditions have been extensively examined [6,37,65,66]. Many previous studies [32,34,65,66] have reported that a high HA increased milk yield. In our study, HA had no effect on herbage intake and milk production, indicating that in a high-quality spring pasture a low HA was sufficient to maintain milk performance, probably as a consequence of similar energy and NDF herbage content between HA. However, low HA resulted in a greater stocking rate (+0.7 cow/ha), which represents an increased milk output per hectare of 23.3%, likely being due to higher levels of herbage utilization [37].

Milk fat concentration and production were not different between HAs and agree with the findings of Kennedy et al. [67] who tested HAs of 13, 16, and 19 kg DM/cow/day (measured at 4 cm above ground level) in early lactation dairy cows. One explanation for this could be the similar herbage intake and thus consumption of fiber, related to the similar NDF content between HAs (averaging 39%) [68] and owing to the proportion of pseudo stem and dead material through the grazing session not varying between HAs as reported by Merino et al. [37].

A decrease in milk fat production and concentration is expected when cows grazed on high-quality pastures supplemented with a high concentrate level (upon 6 kg DM/day) [1,55,69], related to a decrease in ruminal pH [70]. Thus, the equal and low levels of supplements offered in this study, being less than 20% of the total DMI, could be insufficient to decrease the concentration and yield of milk fat [71,72]. Additionally, the lack of effect on milk fat yield could be because NDF in both diets would not have been limiting for rumen function [33].

As herbage DMI was not decreased by HA restriction and the ME and NDF content in the grazed forage did not differ in offered pasture [73] and therefore, the amount of energy supply [74], microbial protein synthesis in the rumen was probably not decreased and thus milk protein production was unaffected by HA, being consistent with the results of Kennedy et al. [29].

In the present study, the supplement offered had no effect on milk protein production, which agrees with O’Mara et al. [75] who reported no significant differences in milk protein production when cows were supplemented with wheat or maize grain. In grazing dairy cows, the starch carbohydrate supplemented should have a similar rate and extent of degradation in the rumen to that of herbage CP in order to maximize N assimilation and to improve milk production and composition [76]. Considering the moderate levels of supplementation offered (around 20% of the total DMI) and because supplements were only offered during the milking time, this response may be due to the herbage was the main source of energy for microbial protein synthesis in the rumen, and therefore, milk protein production was more reliance on herbage intake than on the type of supplemented [6].

Ruminal conditions could be improved by manipulating the starch digestive site through grain processing. Studies reported by Soriano et al. [77] and Alvarez et al. [78], evaluated the effect of processed maize grain on DMI, milk production, and composition of grazing dairy cows, without differences between the different forms of corn treatments. This lack of response to processed maize grains has been related to changes only in the location of the digestion (i.e., more energy available in the rumen with processed grains vs. more energy available post-ruminally with unprocessed grains) without affecting the total energy intake [6].

Conversely, supplementation with CW increased protein concentration by 1.19 g/kg, however, the daily production of protein did not change, as the milk production was unaffected by the supplement type. A greater milk protein concentration could be explained by differences in DM concentrations between HMM (692 g/kg DM) and CW (840 g/kg DM) and by the faster degradability of the carbohydrates of CW than HMM, related to the structure and higher density of maize grain. Thus, the energy of CW could synchronize better with the highly degradable protein of the herbage, improving microbial protein synthesis compared to HMM [79].

Liveweight was more dependent on the level of pasture offered than the type of energy supplement used. Offering a higher HA slightly decreased mean bodyweight by 3% compared to low HA. It is possible that some of the animals grazing high HA, particularly in early lactation, had to mobilize a greater proportion of their body reserves as a result of a higher energy restriction than at low HA. It is well established that cows select green leaf in preference to stem and dead material [80]. Additionally, the mean HA does not reflect the variation in the actual sward height and the height at which the cow is preferentially consuming from (Pulido and Leaver, [64]). Therefore, cows grazing under the high HA conditions might have consumed more green lamina from the upper strata of the pasture, increasing protein intake, which is in accordance with the higher milk and plasma urea observed at a high HA. A high urea concentration in milk and plasma (the latter was above the reference limit of <7.0 mmol L^−1^ recommended by Wittwer et al. [81]), suggests an asynchrony between energy and protein in the rumen [82], and a large amount of ammonia converted into urea in the liver [83]. In addition, the slightly lower LW in cows grazing the high HA can be related to the predisposition of the cows to improve production which could be linked to the numerically greater production in this group, agreeing with the report of Perez-Prieto et al. [59]. The absence of an effect of HA on BCS is in accordance with the results observed on all energy metabolism indicators and agree with the results of Kennedy et al. [29], who offered an HA of 13 kg and 16 kg of DM/cow per day to cows in early lactation supplemented with 4 kg of DM of concentrate.

### 4.3. Metabolic Response

Temperate pastures typically show a high CP and low WSC content, especially in vegetative states of growth during the spring season [84,85,86,87]. Accordingly, it is expected that urea concentrations in milk are to be above 7.0 mmol/L, particularly when supplementation is moderate or low for high-yielding dairy cows [88]. The results of milk urea obtained in this study were thus close to this limit value.

Milk urea concentration was increased by 0.6 mmol/L when a high HA was offered. Regarding urea concentration in plasma, this was increased by 0.89 mmol/L in late lactation dairy cows grazing at a high HA, which was consistent with the increment in milk urea compared to animals offered a low HA. This result could indicate a high rumen ammonia concentration and an excess of ammonia passing into the blood flood and subsequently being converted to urea by the liver [74]. This may be related to a greater intake of forage more digestible and richer in CP, even if no effect on herbage DMI was found. High HA decreases the level of competition between animals increasing the opportunity to be more selective within the grazed horizon [89]. Thus, the pasture selected typically had a higher proportion of green leaf and a lower proportion of stem and dead material, and therefore the herbage eaten has higher digestibility and CP content than at a low HA [90]. In addition, these results can be an indication of an excess of ruminally degradable protein in pastures grazed at low HA that cannot be converted to microbial protein due to insufficient fermentable metabolizable energy in the diet [91], or by asynchrony between the release of protein and energy in the rumen [92], resulting in a lower utilization of dietary protein by rumen microorganisms in cows grazing at low HAs compared to those grazing at high HAs.

It is well known that concentrate supplementation reduces urea concentrations regardless of the HA [33]. Urea concentrations both in milk and plasma were lower for cows supplemented with HMM than those supplemented with CW, however, all were within the reference limits [81]. For dairy cows, this type of HMM presents advantages over other forms of maize grain processing. This is mainly due to the starch of HMM presenting internal structure characteristics which confer a higher ruminal degradability compared to other forms of corn processing [93,94,95], hence it is an excellent supplement to optimize rumen microbial protein synthesis [77].

Plasma urea concentrations for this study were higher than expected for cows on pasture with energy supplement feeds and increased by 0.76 mmol/L when CW was offered. These results suggest that NH_3_-N utilization by rumen microbes was improved when cows were supplemented with a more rumen digestible carbohydrate source (HMM supplement). The slower values of degradability of HMM compared to CW (Dr. Leichtle, personal communication), as a consequence of the maize structure and density, would have synchronized better in time with the high-degradable protein in these pastures consumed through grazing [90]. Bargo et al. [33] reported a plasma urea concentration of 2.08 mmol/L in dairy cows supplemented with maize-based concentrate (1 kg/4 kg of milk) grazing high-quality pasture offered at two herbage allowances (25 and 40 kg DM/day), independent of HA. Delahoy et al. [7] reported a plasma urea concentration of 2.18 mmol/L in cows supplemented with cracked-corn or a steam-flaked (1 kg/4 kg of milk) plus herbage allowance of 40 kg of DM/cow per day in autumn. The high plasma urea observed in our study indicates a low degree of ruminal synchronization of the rate of release of energy and protein, associated with a high content in the pasture of degradable protein, higher than the optimum suggested by Hoover and Stokes [96], and a low content of energy in the diet [92].

The β-hydroxybutyrate, NEFA, and albumin concentrations were similar between HA and ST and within the reference range, indicating an adequate nutritive balance between treatment [97].

### 4.4. Ruminal Fermentation

Ruminal pH and NH_3_-N concentration were not affected by HA nor ST. Ruminal pH values were within the expected values for dairy cows grazing permanent pasture in a vegetative state [98]. As reported by Wales and Doyle [99], ruminal pH changed throughout the day with a maximum value of 6.4 at 06:30 and decreased slowly to the lowest value of 5.28 near to 22:00. We observed a progressive reduction in the pH during the day, which could be explained by a progressive accumulation of WSC in the herbage as the day progresses (in the afternoon) [100], in addition to a reduced rumination (less saliva produced). This pattern arises from the difference in the herbage intake rate and grazing activity throughout the day; these variables are greater during daylight hours (daylight grazing time 72% and night-time grazing time 28% [61]) compared to morning grazing, in preparation for the hours of darkness when little grazing usually occurs. Thus, resulting in the lower pH later in the day.

Van Vuuren et al. [101] found the lowest ruminal pH values 8 h after grazing lactating cows were supplemented with a high or low starch concentrate, which did not occur in this experiment. Those authors suggest that the low ruminal pH may have been affected more by forage fermentation rather than starch fermentation. The former hypothesis agrees with that observed in this experiment regarding the low level of supplementation offered to late lactation dairy cows, and therefore the oscillations of pH were mainly the result of the fermentation of pasture with low concentrations of NDF (average 34.3% in diet) as herbage was in a vegetative stage, the balance between the production and removal rates of acidic products and the buffering capacity [102]. The low pH observed in the afternoon also would be explained by an accumulation of WSC in the herbage as the day progresses (in the afternoon) [101] added to the lack of rumination (less saliva produced) during this time because grazing time mainly occurs during daylight hours (daylight grazing time 72% and night-time grazing time 28%) [61].

Ruminal NH_3_-N concentrations were within the lower limit considered for optimal ruminal physiological performance (8.8 to 17.6 mmol/L [103]) with no significant difference between the HA or ST. This can be explained by the high quality of the pasture with a moderate CP concentration (20.8% of DM) and because all the cows were receiving an equal amount of energetic supplementation. In addition, the ruminal NH_3_-N concentration reduction could be associated with a higher capture of NH_3_-N from the highly ruminal degradable CP of forage, but also with a reduction in total CP intake because energy supplements dilute the CP concentration of the diet [104]. The diurnal variations of NH_3_-N ruminal concentrations were similar for HA and ST, showing a great increase during the afternoon reaching a higher value around 20:00. The former can be explained by the greater herbage intake observed during this period of the day similar to that reported by Bargo et al. [102] and where the effects of the energy supplement in the rumen mainly have ended.

In grazing dairy cows, VFA concentration in ruminal fluid can range between values of 50–150 mmol/L depending on the diet [105]. In our study, the average total VFA production was 92.0 mmol/L, a value in the middle of the range reported by Holmes et al. [105], probably as a result of the lack of synchrony between the CP and energy of the diet. However, the low rumen pH did not agree with the low concentration of VFA, likely because the pH was more related to the reduced rumination and fiber than the nutritive characteristic of the diets. There was a tendency (*p* = 0.086) for a greater concentration of VFA when cows received HMM. Supplementation with cereal-based concentrates high in starch, is characterized by modifying molar proportions of volatile fatty acid, enhancing propionate concentration, therefore, positively influencing the energetic metabolism status of the cow [6,106]. The individual concentrations of acetic, butyric, and propionic acids were not modified by the effect of increasing HA, concurring with a similar DM intake in both groups. On the other hand, the supplementation type also did not change the concentrations of acetic, propionic, and butyric acids.

## 5. Conclusions

In dairy cows grazing on a high-quality spring pasture, decreasing herbage allowance from 30 to 20 kg DM/cow per day does not modify DMI and milk production of dairy cows but increases the efficiency of pasture utilization and milk output per hectare by 23.3%. In addition, HA does not affect grazing behaviors, metabolic response, nor rumen fermentation indicators. Under such conditions, a daily herbage allowance of 20 kg DM/cow per day would help to optimize feed efficiency, profitability, and sustainability of dairy systems based on grazing Considering the low level of supplements used in this study, supplementing grazing dairy cows with either high moisture maize or cracked wheat does not vary grazing behaviors, ruminal fermentation indicators, nor milk production. However, cracked wheat increases milk protein concentration but increases urea in both milk and plasma, evidencing a lack of synchrony between energy and protein supply in the diet.

## Figures and Tables

**Figure 1 animals-11-00919-f001:**
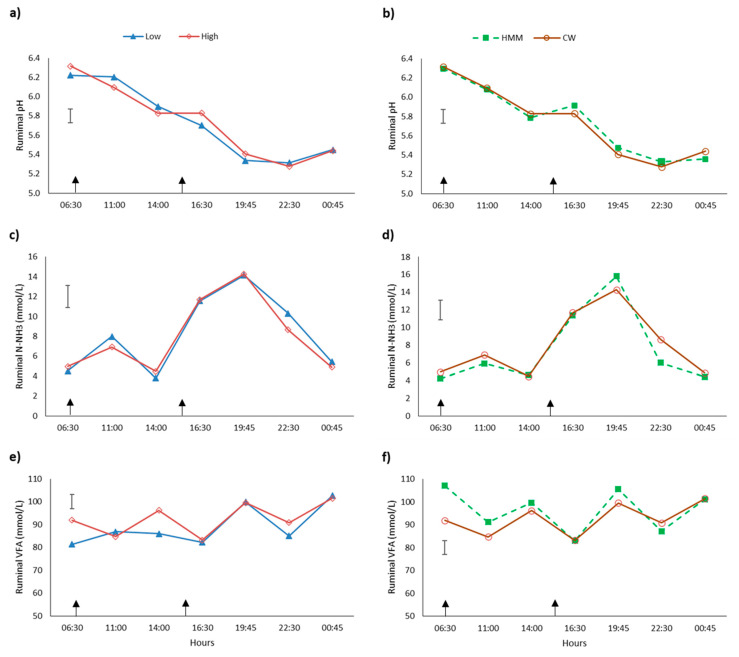
Diurnal variations of pH (**a**,**b**), NH_3_-N (**c**,**d**) and volatile fatty acids (VFA) (**e**,**f**) ruminal concentration in grazing dairy cows. Graphs (**a**,**c**,**e**) shows the two different herbage allowances; Low: 20 kg dry matter/cow/day (▲), or High: 30 kg dry matter/cow/day (◊). Graphs (**b**,**d**,**f**) shows the two different supplement types: high moisture maize HMM (O) or cracked wheat (■) (CW). Data are means drawn from four measurement periods (*n* = 4). Vertical bars represent standard error of the means. Arrows represent supplementation time during milking (06:30 and 15:30 h).

**Table 1 animals-11-00919-t001:** Nutritional composition of pasture (>4 cm, *n* = 9) and supplements (*n* = 5) offered throughout the study. Values are mean ± SD.

	HA ^1^				ST ^2^			
	20 kg	30 kg			HMM		CW	
Component ^3^	X ±	X ±	SD ^4^	*p*-Value	X ±	SD	X ±	SD
DM (%)	18.8	18.8	2.12	0.791	69.2	2.6	84.0	0.05
CP (%)	20.7	20.8	2.01	0.883	7.7	0.51	12.4	0.4
NDF (%)	40.2	39.2	2.38	0.901	10.6	0.09	14.4	1.7
ADF (%)	25.2	24.1	1.71	0.872	1.9	0.35	3.4	0.24
Total Ash (%)	8.1	8.6	0.77	0.908	1.5	0.25	1.4	0.11
ME (MJ/kg DM)	11.54	11.67	0.24	0.752	13.81	0.13	13.31	0.17

^1^ HA = herbage allowance (kg DM/cow/day). ^2^ ST = supplemented type; HMM = high moisture maize; CW = cracked wheat. ^3^ DM: dry matter; CP: crude protein; NDF: neutral detergent fibre; ADF: acid detergent fibre; ME: metabolizable energy. ^4^ SD = standard deviation.

**Table 2 animals-11-00919-t002:** Effect of herbage allowance and supplement type on grazing behavior and dry matter intake of lactating dairy cows (*n* = 16).

	HA ^1^		ST ^2^		SD ^3^	*p*-Value				
Variable	20 kg	30 kg	HMM	CW		HA	ST	HA × ST	LS ^4^	PN ^4^
Grazing Behavior										
Grazing time (min)	446.5	453.9	449.1	451.3	41.94	0.493	0.829	0.064	0.064	0.842
Ruminating time (min)	492.6	481.4	486.6	487.4	42.88	0.302	0.936	0.244	0.0001	0.683
Idling time (min)	501.3	504.7	503.7	502.3	50.06	0.781	0.907	0.502	0.0001	0.616
Dry Matter Intake										
Herbage intake (kg)	14.13	14.20	14.07	14.27	1.477	0.857	0.598	0.444	0.025	0.008
Total DM intake (kg)	17.64	17.70	17.57	17.77	1.477	0.857	0.598	0.444	0.025	0.008
Rate of herbage intake (g DM/min)	31.92	31.56	31.53	31.95	4.308	0.741	0.706	0.541	0.686	0.073

^1^ HA = herbage allowance (kg DM/cow/day). ^2^ ST = supplemented type; HMM = high moisture maize; CW = cracked wheat. ^3^ SD = standard deviation. ^4^ LS = lactation stage (early vs. late); PN = parity number (primiparous vs. multiparous).

**Table 3 animals-11-00919-t003:** Effect of herbage allowance and supplement type level on animal performance and metabolic indicators in the blood of lactating dairy cows (*n* = 16).

	HA ^1^		ST ^2^		SD ^4^	*p*-Value				
Variable	20 kg	30 kg	HMM	CW		HA	ST	HA × ST	LS ^5^	PN ^5^
Milk Production and Composition										
^3^ Milk production (kg/day)	22.95	24.04	23.98	23.01	2.883	0.145	0.196	0.352	0.0001	0.057
^3^ Milk fat production (g/day)	811.97	833.49	835.66	809.79	168.147	0.636	0.569	0.106	0.089	0.108
^3^ Milk protein production (g/day)	783.37	817.03	805.70	794.69	97.835	0.197	0.675	0.777	0.0001	0.009
^3^ 4% FCM production (kg/day)	21.44	22.21	22.24	21.40	3.153	0.377	0.318	0.089	0.0003	0.050
Milk fat concentration (g/kg)	35.34	35.25	34.82	35.78	6.688	0.955	0.583	0.390	0.017	0.816
Milk protein concentration (g/kg)	33.98	34.04	33.42	34.60	2.289	0.916	0.045	0.084	0.011	0.223
Bodyweight and Body Condition Score										
^3^ Bodyweight (kg)	551.2	533.8	543.3	541.6	24.68	0.009	0.804	0.531	0.004	0.0001
^3^ Body condition score (1–5)	3.00	2.98	2.95	3.03	0.257	0.727	0.295	0.512	0.0001	0.053
Metabolism Indicators										
Milk urea (mmol/L)	6.60	7.23	6.56	7.28	0.857	0.005	0.002	0.245	0.068	0.025
Plasma urea (mmol/L)	6.62	7.51	6.69	7.44	1.352	0.010	0.029	0.502	0.755	0.981
Plasma β-hydroxybutyrate (mmol/L)	0.65	0.64	0.68	0.62	0.124	0.782	0.054	0.583	0.161	0.067
Plasma NEFA (mmol/L)	0.449	0.448	0.439	0.459	0.1963	0.976	0.688	0.305	0.004	0.016
Plasma albumin (g/L)	41.63	41.79	41.48	41.93	3.474	0.853	0.614	0.890	0.131	0.730

^1^ HA = herbage allowance (kg DM/cow/day). ^2^ ST = supplemented type; HMM = high moisture maize; CW = cracked wheat. ^3^ Means adjusted by covariates. ^4^ SD = standard deviation. ^5^ LS = lactation stage (early vs. late); PN = parity number (primiparous vs. multiparous). None of the interactions were significant except for LS × HA in milk urea.

**Table 4 animals-11-00919-t004:** Effect of herbage allowance and supplement type on ruminal functions of grazing dairy cows (*n* = 4).

	HA ^1^		ST ^2^		SEM ^3^	*p*-Value		
Variable	20 kg	30 kg	HMM	CW		HA	ST	HA × ST
Mean ruminal pH	5.73	5.75	5.74	5.74	0.072	0.810	0.856	0.528
NH_3-_N (mmol/L)	8.25	7.71	7.49	8.47	1.125	0.650	0.416	0.728
Total VFA (mmol/L)	90.21	96.66	96.28	90.59	3.055	0.079	0.230	0.132
VFA (molar %)								
Acetic: total VFA (%)	61.65	62.56	61.70	62.51	1.695	0.610	0.650	0.762
Propionic: total VFA (%)	21.41	21.27	21.74	20.94	1.146	0.909	0.509	0.921
Butyric: total VFA (%)	12.90	12.37	12.59	12.68	0.565	0.388	0.872	0.528
Acetic: propionic ratio	3.01	3.05	2.96	3.10	0.197	0.839	0.490	0.896

^1^ HA = herbage allowance (kg DM/cow/day). ^2^ ST = supplemented type; HMM = high moisture maize; CW = cracked wheat. ^3^ SEM = standard error of the mean.

## Data Availability

The data that support the findings of this study are available at https://figshare.com/s/c45c33f3a6e4f3142440 (public DOI: 10.6084/m9.figshare.13818920) (accessed on 22 March 2020).

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
