# Peer review of "Metabolic and Productive Response and Grazing Behavior of Lactating Dairy Cows Supplemented with High Moisture Maize or Cracked Wheat Grazing at Two Herbage Allowances in Spring"

_animals, 2021, doi:10.3390/ani11040919_

Round 1
Reviewer 1 Report
The paper “Metabolic and productive response and grazing behaviour of lactating dairy cows supplemented with high moisture maize or cracked wheat grazing at two herbage allowance in spring by Verónica M. Merino et al. represents an impressive work, extremely complex and complete, on the relations between diet, digestive physiology, metabolism and milk yield and composition. Nevertheless, it needs to be improved from many points of view.
Introduction: -
Lines 54-58, the two aspects (milk per ha and better efficiency) are not in contradiction.
Materials and Methods: -
Lines 135-137, the milking times are different respect to line 218; moreover, please to explain why 50% of cows were in late lactation (no energy deficiency) and why the results have not been discussed in this respect.
Lines 151-154, after grazing the residue remained on the soil? And how long it takes for next grazing?
Lines 169-172, the amounts of Cu and particularly of iodine are extremely high, are you sure of them?
Lines 220-222, the big difference in milking intervals is usually cause of some differences in fat contents, besides yields; were the pools made according to the yields of each milking?
Line 232, the centrifuge speed (480 is expressed as g-gravity? Or number of revolutions? So low?)
Lines 235-245, the cannulated cows were only in late lactation, but this was never commented
Results: -
Any possibility to evaluate the effect of lactation stage?
Figure 1, it could be useful to indicate what are the left and right graphs, besides the meaning of arrows
Line 346, perhaps (0,054)?
Discussion:-
Lines 379-381, the sentence is not clear to me, please to verify;
Lines 394-395, the authors wrote: “as evidenced by lower post-grazing herbage mass (-127 kg DM/ha) and post-grazing 394 sward height (-0.7 cm)” please to better explain -127 and -0,7…respect to what?
Lines 422-427, a further reason of the lack of effect of concentrate could be the low genetic merit, as well as the fact that 50% of cows were in late lactation.
Lines 483-484, the authors wrote: “protein concentration by 1.19 g/day”, but concentration cannot be g/day ;
Lines 485-489, how can different DM concentration affect fermentability? Perhaps starch structure or density of particles. Please to explain. Moreover, the ammonia behaviour in rumen is similar.
Line 490, Metabolic response…urea diffusion in body compartments (including mammary gland) is quick and complete; therefore, its concentration in plasma and milk is very similar and must be discussed together:
Lines 495-509, I see several contradictions: urea is higher in 30 HA and CW, but rumen ammonia is similar as well as DMI and protein contents of feeds; thus many interpretation attempts seems speculations
Line 506, compared….compared…
Lines 531-533, cows of low genetic merit, after 3rd month and “properly” fed cannot mobilize fat (what about BCS changes?). For protein balance albumins are without meaning (except for serious deficiency);
Lines 535-549, again, I see some contradictions: the pH is reduced in late afternoon-night, but VFA concentration seems almost constant, while ammonia is high (in part of time). The authors must consider this, besides the higher forage intake in the afternoon;
Lines 572-575, very strange interpretation that C4 could be produced by bacteria during storage of a material with a 16% water content.
Conclusions:
I suggest to authors to modify conclusions according to the suggested changes in the text (particularly for line 589). Furthermore, it would be better emphasize that in case of low genetic merit cows (better adapted for grazing system) and good sward quality, the best goal is efficient grazing and maximum milk per hectare.
The paper “Metabolic and productive response and grazing behaviour of lactating dairy cows supplemented with high moisture maize or cracked wheat grazing at two herbage allowance in spring by Verónica M. Merino et al. represents an impressive work, extremely complex and complete, on the relations between diet, digestive physiology, metabolism and milk yield and composition. Nevertheless, it needs to be improved from many points of view.
Introduction: -
Lines 54-58, the two aspects (milk per ha and better efficiency) are not in contradiction.
Materials and Methods: -
Lines 135-137, the milking times are different respect to line 218; moreover, please to explain why 50% of cows were in late lactation (no energy deficiency) and why the results have not been discussed in this respect.
Lines 151-154, after grazing the residue remained on the soil? And how long it takes for next grazing?
Lines 169-172, the amounts of Cu and particularly of iodine are extremely high, are you sure of them?
Lines 220-222, the big difference in milking intervals is usually cause of some differences in fat contents, besides yields; were the pools made according to the yields of each milking?
Line 232, the centrifuge speed (480 is expressed as g-gravity? Or number of revolutions? So low?)
Lines 235-245, the cannulated cows were only in late lactation, but this was never commented
Results: -
Any possibility to evaluate the effect of lactation stage?
Figure 1, it could be useful to indicate what are the left and right graphs, besides the meaning of arrows
Line 346, perhaps (0,054)?
Discussion:-
Lines 379-381, the sentence is not clear to me, please to verify;
Lines 394-395, the authors wrote: “as evidenced by lower post-grazing herbage mass (-127 kg DM/ha) and post-grazing 394 sward height (-0.7 cm)” please to better explain -127 and -0,7…respect to what?
Lines 422-427, a further reason of the lack of effect of concentrate could be the low genetic merit, as well as the fact that 50% of cows were in late lactation.
Lines 483-484, the authors wrote: “protein concentration by 1.19 g/day”, but concentration cannot be g/day ;
Lines 485-489, how can different DM concentration affect fermentability? Perhaps starch structure or density of particles. Please to explain. Moreover, the ammonia behaviour in rumen is similar.
Line 490, Metabolic response…urea diffusion in body compartments (including mammary gland) is quick and complete; therefore, its concentration in plasma and milk is very similar and must be discussed together:
Lines 495-509, I see several contradictions: urea is higher in 30 HA and CW, but rumen ammonia is similar as well as DMI and protein contents of feeds; thus many interpretation attempts seems speculations
Line 506, compared….compared…
Lines 531-533, cows of low genetic merit, after 3rd month and “properly” fed cannot mobilize fat (what about BCS changes?). For protein balance albumins are without meaning (except for serious deficiency);
Lines 535-549, again, I see some contradictions: the pH is reduced in late afternoon-night, but VFA concentration seems almost constant, while ammonia is high (in part of time). The authors must consider this, besides the higher forage intake in the afternoon;
Lines 572-575, very strange interpretation that C4 could be produced by bacteria during storage of a material with a 16% water content.
Conclusions:
I suggest to authors to modify conclusions according to the suggested changes in the text (particularly for line 589). Furthermore, it would be better emphasize that in case of low genetic merit cows (better adapted for grazing system) and good sward quality, the best goal is efficient grazing and maximum milk per hectare.
Reviewer 2 Report
Congratulations, a good job. Sized and explained. Below I propose some annotations in order to improve the final version.
General comments: Regarding the representation of the results and their future discussion, I think the nutrition information would improve if it were in a table. Correctly explained. of the pastures, as well as of the concentrate.
Specific comments:
L 30. HA is not defined in the abstract.
L 35. DM is not defined, please define it in L30.
L 40. DMI is not defined in the abstract.
Question. What is the sample numbers for each variable? It appears different SD. Is the same in all treatments?
L338. the title of the table is sometimes justified, sometimes not. Please check the format.
L 339. I suppose you do not put letters since significant differences are only observed within treatments and, as there are only two options, it is understood that they are different. Is that so?
L 339. In the case, for example, of the urea concentration in milk, if the standard errors are equal, the HMM measurement should be different from the HA with 30 kg, these differences would not be interesting for the discussion?
L 367. Mean +- SD? confinance interval?
L 367. Please, Specify each section. (a, b, c…)
L 367. What is the meaning of the arrows?
L 449. “the high urea concentration observed in milk was consistent with the urea…, biologically, wouldn't it be more logical for it to be the other way around?
L 518. Don't you think it would be interesting to go deeper into the quality of the protein and not just the amount of protein?
L 580. There are several comments where it says that it does not improve production rates but increases metabolites related to the "waste" of protein, and this could be related to a question of an amino acid mismatch. In my opinion it should be compared to some work with bypass protein (for example)
